# Semi-Synthesis and In Vitro Anti-Cancer Evaluation of Magnolol Derivatives

**DOI:** 10.3390/molecules26144302

**Published:** 2021-07-15

**Authors:** Xiao-Long Sun, Mei-Lin Zhu, Yi-Qun Dai, Hong-Mei Li, Bo-Han Li, Hui Ma, Chang-Hao Zhang, Cheng-Zhu Wu

**Affiliations:** 1School of Pharmacy, Bengbu Medical College, 2600 Donghai Road, Bengbu 233030, China; sunxiaolong96@foxmail.com (X.-L.S.); zlyk521@126.com (M.-L.Z.); daiyiqun25@126.com (Y.-Q.D.); athongmei@bbmc.edu.cn (H.-M.L.); libohan1228@foxmail.com (B.-H.L.); mahui9513@foxmail.com (H.M.); 2Key Laboratory of Natural Medicines of the Changbai Mountain, College of Pharmacy, Yanbian University, Ministry of Education, 977 Gongyuan Road, Yanji 133002, China

**Keywords:** magnolol, semi-synthesis, cytotoxicity, migration, invasion

## Abstract

Magnolol (MAG), a biphenolic neolignan, has various biological activities including antitumor effects. In this study, 15 MAG derivatives were semi-synthesized and evaluated for their in vitro anticancer activities. From these derivatives, compound **6a** exhibited the best cytotoxic activity against four human cancer cell lines, with IC_50_ values ranging from 20.43 to 28.27 μM. Wound-healing and transwell assays showed that compound **6a** significantly inhibited the migration and invasion of MDA-MB-231 cells. In addition, Western blotting experiments, performed using various concentrations of **6a**, demonstrated that it downregulates the expression of HIF-1α, MMP-2, and MMP-9 in a concentration-dependent manner. Overall, these results suggest that substituting a benzyl group having F atoms substituted at the C2 position on MAG is a viable strategy for the structural optimization of MAG derivatives as anticancer agents.

## 1. Introduction

Triple-negative breast cancer (TNBC) is defined by a lack of expression of the estrogen receptor, epidermal growth factor receptor-2, and progesterone receptor-2. TNBC shows a higher rate of distant recurrence and metastasis, as well as a poorer prognosis, than other breast cancer subtypes [1,2]. Moreover, TNBC lacks standardized treatment guidelines and effective drugs [3]. For these reasons, anti-TNBC agents with high efficiency and low toxicity are needed. In contrast to chemotherapy, natural products have many advantages, such as low toxicity and side effects, multiple targets, and reversal of cancer-drug resistance [4,5].

Natural products play an important role in the research and development of new drugs [6,7]. Natural products can have many advantages over chemotherapy, such as low toxicity, low side effects, multiple targets, and reversal of resistance to cancer drugs. Magnolol (MAG, 5,5′-diallyl-2,2′-dihydroxybiphenyl), a lignan isolated mainly from *Magnolia officinalis*, Rehder & E.H. Wilson (“Hou Po” in Chinese) [8], exhibits a huge range of biological activities such as anti-inflammatory [9], neuroprotective [10], antibacterial [11], antioxidant [12], and antitumor [13] effects. In addition, many recent studies have shown that MAG can inhibit the proliferation, migration, metastasis, and angiogenesis of various cancer cell lines [14,15,16,17], suggesting that MAG has potential as a lead compound for the discovery of anticancer agents.

Previous studies have shown that a series of MAG derivatives showed better cytotoxic activity than MAG itself [18,19,20,21,22]. These results indicated that free phenolic hydroxyl groups and hydrophobic side chains are the necessary active groups for magnolol to exert antitumor effects. However, in a previous study we showed that monosubstitution of a benzyl group with Cl atoms at the phenolic hydroxyl group of honokiol significantly improved the cytotoxic activity of honokiol against cancer cells [23]. In this study, we modified the phenolic hydroxyl of MAG to obtain a series of novel MAG derivatives and examined the cytotoxic activity of these derivatives against four human cancer cell lines, as well as their inhibition of the migration and invasion of MDA-MB-231 cells.

## 2. Results and Discussion

### 2.1. Synthesis and Characterization of the Derivatives

A series of 15 MAG derivatives were designed and synthesized using MAG as the starting material (Scheme 1.). Methyl, ethyl, benzyl, 2-fluorobenzyl, 3-fluorobenzyl, 4-fluorobenzyl, 2-chlorobenzyl, 3-chlorobenzyl, and 4-chlorobenzyl were substituted on the phenol groups of MAG via the Williamson reaction to obtain monosubstituted products **1a**–**9a** and disubstituted derivatives **1b**–**6b**. The chemical structures of the MAG derivatives were characterized using ^1^H-NMR, ^13^C-NMR, ESI-MS, and HR-ESI-MS. (see Appendix A).

### 2.2. In Vitro Cytotoxic Activity

All synthesized derivatives were evaluated for their in vitro cytotoxic activity against four human cancer cell lines using the MTT assay with Taxol as a positive control. The inhibited cell growth to 50% of the control (IC_50_) was determined for the compounds, and the results are summarized in Table 1. Compounds **1a***–***9a** all exhibited cytotoxic activity against one or more types of cancer cell lines. Compounds **3a**, **4a**, **5a**, and **6a** had better cytotoxic activity against all four human cancer cell lines than compounds **2a** or **7a***–***9a**, with IC_50_ values ranging from 20.43 to 30.21 μM. Compound **6a** showed the best cytotoxic activity against the human TNBC MDA-MB-231 cell line with an IC_50_ value of 20.43 μM, which indicates **6a** was approximately 2.5 times stronger than magnolol.

The preliminary structure–activity relationship (SAR) study suggested that *O*-alkylation of the 2-OH or 2′-OH group of MAG improved cytotoxic activity against MCF-7, CNE-2Z, and SMMC-7721 cells, while both 2-OH and 2′-OH alkylated derivatives were inactive at 50 μM. Therefore, the hydroxyl group of MAG may be one of the active groups for cytotoxic activity. Moreover, the substituents on the benzyl ring had a significant influence on the cytotoxic activity of the derivatives for MDA-MB-231 cells. The activities of the F-substituted derivatives (**4a**, **5a**, and **6a**) were stronger than those of the Cl-substituted derivatives. Compound **3a**, which does not bear any substitution on the benzene ring, showed good cytotoxic activity against MDA-MB-231 cells.

### 2.3. Preliminary Screening of the Inhibitory Effect on MDA-MB-231 Cell Migration

Tumor cell migration is an important event in the invasion and metastatic cascade of cancer [24]. In the preliminary screening, wound-healing assays indicated that compounds **5a**, **6a**, and **9a** exhibited significant inhibitory effects on MDA-MB-231 cell migration at a concentration of 10 mm (Figure 1). As shown in Figure 2, transwell assays indicated that compounds **2a**–**6a**, **8a**, and **9a** also showed potent inhibitory activities on the migration of MDA-MB-231 cells compared with MAG. Preliminary screening results showed that compound **6a** had the best anti-migration activity of the MAG derivatives. Therefore, compound **6a** was selected for further experiments.

### 2.4. Anti-Proliferative Activity of Compound ***6a*** in MDA-MB-231 Cells

To understand the anticancer activity of **6a**, MDA-MB-231 cells were treated with increasing concentrations of the compound (0, 10, 20, 30, 40, and 50 μM) for 24, 48, and 72 h, and cell proliferation was measured using an MTT assay. As shown in Figure 3, **6a** showed significant antiproliferative activity on MDA-MB-231 cells in a time and concentration-dependent manner. When the concentration of **6a** was increased to 50 μM, cell proliferation was almost completely inhibited. Therefore, 5, 10, and 20 μM **6a** were used in subsequent experiments.

### 2.5. Suppressed Migration and Invasion of MDA-MB-231 Cells by Compound ***6a***

Tumor invasion and metastasis are the causes of poor prognosis and survival in patients with breast cancer [25,26]. Therefore, controlling invasion and metastasis is an important therapeutic strategy for TNBC [27]. We examined the effects of compound **6a** on human MDA-MB-231 TNBC cells. The migration and invasion of MDA-MB-231 cells decreased as the concentration of **6a** increased, indicating that **6a** significantly suppressed the migration and invasion of the cells in a concentration-dependent manner (Figure 4A,B).

Hypoxia is a common feature in many solid tumors that promotes cancer invasion, metastasis, epithelial-mesenchymal transition, and angiogenesis via hypoxia inducible factor-1α (HIF-1α) [28,29,30,31]. This key hypoxic regulator is highly expressed in breast cancer and facilitates tumor migration and invasion through matrix metalloproteinases (MMPs) [32]. Western blotting showed that treatment with **6a** significantly downregulated HIF-1α and its downstream proteins MMP-2 and MMP-9 (Figure 4C). These results indicated that **6a** suppressed the migration and invasion of MDA-MB-231 cells regulated by HIF-1α.

In our previous work, we found that the benzyl group substituted with two Cl atoms at phenolic hydroxyl group of honokiol, the isomer of MAG, can significantly improve the anti-proliferative activity [23]. In this study, we found that the 2-OH or 2′-OH of MAG replaced by Cl-substituted benzyl group can also improve the anti-proliferative activity (30 μM vs. 46 μM against MCF-7). Moreover, several studies have shown that selective introduction of fluorine elements into a therapeutic or diagnostic small molecule candidate can improve a number of pharmacokinetic and physicochemical properties such as metabolic stability and membrane permeation [33,34,35]. Herein, the synthesis of MAG derivatives with F-substituted benzyl group further confirmed that with greater performance in anti-proliferative, anti-migratory and anti-invasive experiments than the Cl-substituted derivatives; and the anti-proliferative activity of products with the phenolic hydroxyl group replaced by benzyl group substituted with two F atoms will be expected.

## 3. Materials and Methods

### 3.1. General Information

All reagents were commercially available and used without further purification. MAG was purchased from Xinmingtai Chemical Co., Ltd. (Wuhan, China). All reactions were monitored by thin-layer chromatography on silica gel F_254_ (Qindao Haiyang Chemical Co., Qindao, China). The products were purified by column chromatography (300–400 mesh silica gel, Qindao Haiyang Chemical Co., Qindao, China). All NMR spectra were recorded on a Bruker Avance II 600 MHz instrument (Bruker, Billerica, MA, USA) using CDCl_3_ or DMSO-d_6_ as the solvent with TMS as the internal standard. Chemical shifts (δ) were reported in parts per million (ppm) and the coupling constants (J) were given in Hertz. High-resolution electrospray ionization mass spectrometry (HR-ESI-MS) spectra were recorded on a Thermo Scientific LTQ Orbitrap XL mass spectrometer (Bruker, Bremerhaven, Germany) with electrospray ionization. Taxol was purchased from Shanghai yuanye Bio-Technology Co., Ltd. (Shanghai, China, batch numbers: H15F10X80839). Anti-HIF-1α antibodies were purchased from Abcam (Cambridge, UK, batch numbers: GR244245-51). Anti-MMP-9 and MMP-2 antibodies were purchased from Proteintech Group (Chicago, IL, USA, batch numbers of MMP-9: 00074416; batch numbers of MMP-2: 00066968).

### 3.2. General Procedure for the Preparation of ***1a**–**9a*** and ***1b**–**6b***

MAG (80 mg, 0.3 mM) was dissolved in *N*,*N*-dimethylformamide (DMF, 3 mL). Sodium carbonate solution (10%, 2 mL) and the desired halohydrocarbon (0.75 mM) were added and the mixture was stirred for 6 h in an oil bath at 65 °C. The reaction was quenched with water and the crude product extracted with ethyl acetate (EtOAc) three times. The combined extracts were dried over Na_2_SO_4_ and concentrated under reduced pressure. The residue was purified using silica gel (300–400 mesh) column chromatography with a petroleum ether:EtOAc (50:1 to 30:1) eluent to afford compounds **1a**–**9a** and **1b**–**6b**.

*5,5*′*-Diallyl-2-methoxy-1,1*′*-biphenyl-2*′*-ol (***1a***).* Yield: 48.1%, yellow oil, ^1^H-NMR (CD_3_OD, 500 MHz) δ 7.13 (1H, dd, *J* = 7.0, 1.9 Hz, H-4), 7.00 (1H, d, *J* = 1.9 Hz, H-3), 6.97 (1H, dd, *J* = 6.9, 2.0 Hz, H-4′), 6.96 (1H, d, *J* = 6.9 Hz, H-6), 6.90 (1H, d, *J* = 1.9 Hz, H-3′), 6.78 (1H, d, *J* = 6.9 Hz, H-6′), 5.96 (2H, m, H-8, H-8′), 5.07 (2H, m, H-9), 5.02 (2H, m, H-9′), 3.74 (3H, s, H-10), 3.34 (2H, d, *J* = 5.6 Hz), 3.30 (2H, d, *J* = 5.3 Hz); ^13^C NMR (CD_3_OD, 125 MHz) δ 155.3 (C-2), 152.4 (C-2′), 138.1 (C-8′), 137.9 (C-8), 132.0 (C-5′), 131.4 (C-5), 131.1 (C-6′), 130.8 (C-6), 128.2 (C-4′), 128.1 (C-4), 127.8 (C-1), 126.1 (C-1′), 115.4 (C-3′), 114.2 (C-9′), 114.0 (C-9), 111.2 (C-3), 54.9 (C-10), 39.0 (C-7, C-7′); ESIMS: *m*/*z* 281.17 [M + H]^+^.

*5,5*′*-Diallyl-2,2*′*-dimethoxy-1,1*′*-biphenyl (***1b***).* Yield: 20.9%, yellow oil, ^1^H-NMR (CD_3_OD, 500 MHz) δ 7.10 (2H, dd, *J* = 7.1, 2.0 Hz, H-4, H-4′), 6.92 (2H, d, *J* = 1.1 Hz, H-6, H-6′), 6.91 (2H, d, *J* = 3.9 Hz, H-3, H-3′), 5.96 (2H, m, H-8, H-8′), 5.03 (4H, m, H-9, H-9′), 3.69 (6H, s, H-10, H-10′), 3.32 (4H, d, *J* = 5.6 Hz, H-7, H-7′); ^13^C NMR (CD_3_OD, 125 MHz): δ 155.5 (C-2, C-2′), 137.9 (C-8, C-8′), 131.6 (C-5, C-5′), 131.1 (C-6, C-6′), 128.1 (C-4, C-4′), 128.0 (C-1, C-1′), 114.2 (C-9, C-9′), 110.9 (C-3, C-3′), 54.8 (C-10, C-10′), 38.9 (C-7, C-7′); ESIMS: *m*/*z* 295.18 [M + H]^+^.

*5,5*′*-Diallyl-2-ethoxy-1,1*′*-biphenyl-2*′*-ol (***2a***).* Yield: 73.6%, yellow oil, ^1^H-NMR (DMSO-*d_6_*, 500 MHz): δ 8.86 (1H, s, –OH), 7.07 (1H, d, *J* = 2.4 Hz, H-6), 7.05 (1H, d, *J* = 2.3 Hz, H-6′), 6.95 (1H, d, *J* = 8.4 Hz, H-3), 6.95 (1H, dd, *J* = 2.3, 2.3 Hz, H-4), 6.90 (1H, dd, *J* = 2.4, 2.3 Hz, H-4′), 6.79 (1H, d, *J* = 8.1 Hz, H-3′), 5.94 (2H, m, H-8, H-8′), 5.05 (4H, m, H-9, H-9′), 3.96 (2H, q, *J* = 7.0 Hz, H-10), 3.30 (4H, d, *J* = 6.8 Hz, H-7, H-7′), 1.17 (3H, t, *J* = 6.9 Hz, H-11); ^13^C NMR (DMSO-*d_6_*, 125 MHz): δ 154.3 (C-2), 152.8 (C-2′), 138.3 (C-8′), 138.1 (C-8), 131.4 (C-5′), 131.3 (C-5), 131.0 (C-6′), 129.2 (C-6), 128.0 (C-4′), 127.9 (C-1, C-4), 125.4 (C-1′), 115.4 (C-9, C-9′), 115.2 (C-3′), 112.8 (C-3), 63.6 (C-10), 38.7 (C-7, C-7′), 14.7 (C-11); ESIMS: *m*/*z* 317.15 [M + Na]^+^.

*5,5*′*-Diallyl-2,2*′*-diethoxy-1,1*′*-biphenyl (***2b***).* Yield: 6.8%, yellow oil, ^1^H-NMR (DMSO-*d_6_*, 500 MHz): δ 7.07 (2H, dd, *J* = 8.3, 2.0 Hz, H-4, H-4′), 6.94 (2H, d, *J* = 8.3 Hz, H-3, H-3′), 6.94 (2H, s, H-6, H-6′), 5.95 (2H, m, H-8, H-8′), 5.05 (4H, m, H-9, H-9′), 3.95 (4H, q, *J* = 7.0 Hz, H-10, H-10′), 3.31 (4H, d, *J* = 6.8 Hz, H-7, H-7′), 1.17 (6H, t, *J* = 6.9 Hz, H-11, H-11′); ^13^C NMR (DMSO-*d_6_*, 125 MHz): δ 154.3 (C-2, C-2′), 138.1 (C-8, C-8′), 131.2 (C-5, C-5′), 130.9 (C-6, C-6′), 128.1 (C-4, C-4′), 127.6 (C-1, C-1′), 115.4 (C-9, C-9′), 112.2 (C-3, C-3′), 63.3 (C-10, C-10′), 38.6 (C-7, C-7′), 14.7 (C-11, C-11′); ESIMS: *m*/*z* 345.18 [M + Na]^+^.

*5,5*′*-Diallyl-2-benzyloxy-1,1*′*-biphenyl-2*′*-ol (***3a***).* Yield: 68.8%, pale yellow oil, ^1^H-NMR (DMSO-*d_6_*, 500 MHz): δ 9.04 (1H, s, –OH), 7.31 (5H, m, Ar-H), 7.08 (1H, m, H-3), 7.02 (2H, m, H-6, H-6′), 6.94 (2H, dd, *J* = 5.7, 2.3 Hz, H-4, H-4′), 6.83 (1H, m, H-3′), 5.93 (2H, m, H-8, H-8′), 5.09 (2H, d, *J* = 17.1 Hz, H-9), 5.03 (2H, s, H-10), 5.00 (2H, d, *J* = 11.0 Hz, H-9′), 3.32 (2H, d, *J* = 6.8 Hz, H-7), 3.26 (2H, d, *J* = 6.7 Hz, H-7′); ^13^C NMR (DMSO-*d_6_*, 125 MHz): δ 154.0 (C-2′), 152.9 (C-2), 138.2 (C-8′), 138.0 (C-8), 137.5 (C-11), 131.4 (C-5′), 131.3 (C-5, C-6′), 129.2 (C-6), 128.1 (C-4, C-13, C-15, C-4′), 127.9 (C-1), 127.4 (C-14), 127.0 (C-12, C-16, C-1′), 115.5 (C-9′), 115.3 (C-3′), 115.2 (C-9), 112.9 (C-3), 69.4 (C-10), 38.7 (C-7, C-7′); HRESIMS: calcd. for C_25_H_24_O_2_Na^+^ [M + Na]^+^ 379.1669, found 379.1673.

*5,5*′*-Diallyl-2,2*′*-bis (benzyloxy)-1,1*′*-biphenyl (***3b***).* Yield: 6.74%, rhombic white crystal; ^1^H-NMR (DMSO-*d_6_*, 500 MHz): δ 7.25 (10H, m, Ar-H), 7.10 (2H, dd, *J* = 8.3, 2.2 Hz, H-4, H-4′), 7.06 (2H, s, H-6, H-6′), 7.05 (2H, d, *J* = 8.4 Hz, H-3, H-3′), 5.96 (2H, m, H-8, H-8′), 5.04 (4H, m, H-9, H-9′), 5.01 (4H, s, H-10, H-10′), 3.32 (4H, d, *J* = 6.8 Hz, H-7, H-7′); ^13^C NMR (DMSO-*d_6_*, 125 MHz): δ 154.0 (C-2, C-2′), 138.0 (C-11, C-11′), 137.4 (C-8, C-8′), 131.4 (C-5, C-5′), 131.3 (C-6, C-6′), 128.3 (C-4, C-4′), 128.2 (C-13, C-15, C-13′, C-15′), 127.5 (C-14, C-14′), 127.4 (C-1, C-1′), 126.9 (C-12, C-16, C-12′, C-16′), 115.5 (C-9, C-9′), 112.8 (C-3, C-3′), 69.4 (C-10, C-10′), 38.7 (C-7, C-7′); HRESIMS: calcd. for C_32_H_30_O_2_Na^+^ [M + H]^+^ 447.2283, found 447.2280.

*5,5*′*-Diallyl-2-((2-fluorobenzyl)oxy)-1,1*′*-biphenyl-2*′*-ol (***4a***).* Yield: 62.3%, pale yellow oils; ^1^H-NMR (DMSO-*d_6_*, 500 MHz): δ 9.03 (1H, s, –OH), 7.39 (1H, m, H-13), 7.34 (1H, m, H-14), 7.18 (1H, m, H-4′), 7.14 (1H, m, H-3), 7.08 (1H, m, H-15), 7.07 (1H, m, H-6), 7.03 (1H, m, H-6′), 6.92 (2H, m, H-4, H-16), 6.81 (1H, m, H-3′), 5.92 (2H, m, H-8, H-8′), 5.09 (2H, m, H-9), 5.08 (2H, s, H-10), 5.03 (2H, m, H-9′), 3.33 (2H, d, *J* = 6.8 Hz, H-7), 3.24 (2H, d, *J* = 6.7 Hz, H-7′); ^13^C NMR (DMSO-*d_6_*, 125 MHz): δ 158.8 (d, ^1^*J*_CF_ = 243.7 Hz, C-12), 153.8 (C-2′), 152.9 (C-2), 138.2 (C-8′, C-8), 138.0 (C-5′), 131.6 (C-5), 131.5 (C-6′), 131.3 (C-6), 129.7 (d, ^3^*J*_CF_ = 4.1 Hz, C-14), 129.6 (C-4′), 129.2 (C-4), 128.1 (d, ^3^*J*_CF_ = 4.9 Hz, C-16), 128.0 (d, ^2^*J*_CF_ = 12.2 Hz, C-11), 125.0 (C-1), 124.4 (C-1′), 124.3 (d, ^4^*J*_CF_ = 3.2 Hz, C-15), 115.5 (C-3′), 115.4 (C-9′), 115.2 (C-9), 115.1 (d, ^2^*J*_CF_ = 20.8 Hz, C-13), 112.9 (C-3), 63.6 (d, ^3^*J*_CF_ = 4.0 Hz, C-10), 38.7 (C-7, C-7′); HRESIMS: calcd. for C_25_H_23_FO_2_Na^+^ [M + Na]^+^ 397.1574, found 397.1574.

*5,5*′*-Diallyl-2,2*′*-bis ((2-fluorobenzyl)oxy)-1,1*′*-biphenyl (***4b***).* Yield: 5.6%, pale yellow oils; ^1^H-NMR (DMSO-*d_6_*, 500 MHz): δ 7.33 (2H, m, H-14, H-14′), 7.25 (2H, m, H-13, H-13′), 7.17 (2H, m, H-16, H-16′), 7.09 (2H, m, H-15, H-15′), 7.08 (4H, m, H-3, H-4, H-3′, H-4′), 7.05 (2H, s, H-6, H-6′), 5.90 (2H, m, H-8, H-8′), 5.03 (4H, m, H-9, H-9′), 5.03 (4H, s, H-10, H-10′), 3.30 (4H, d, *J* = 6.7 Hz, H-7, H-7′); ^13^C NMR (DMSO-*d_6_*, 125 MHz): δ 158.8 (d, ^1^*J*_CF_ = 244.0 Hz, C-12, C-12′), 153.7 (C-2, C-2′), 137.9 (C-8, C-8′), 131.6 (C-5, C-6, C-5′, C-6′), 129.8 (d, ^3^*J*_CF_ = 8.2 Hz, C-16, C-16′), 129.6 (d, ^3^*J*_CF_ = 4.1 Hz, C-14, C-14′), 128.4 (C-4, C-4′), 127.2 (C-1, C-1′), 124.2 (d, ^4^*J*_CF_ = 3.5 Hz, C-15, C-15′), 124.1 (d, ^2^*J*_CF_ = 15.1 Hz, C-11, C-11′), 115.5 (C-9, C-9′), 115.2 (d, ^2^*J*_CF_ = 20.8 Hz, C-13, C-13′), 112.8 (C-3, C-3′), 63.7 (d, ^3^*J*_CF_ = 4.0 Hz, C-10, C-10′), 38.6 (C-7, C-7′); HRESIMS: calcd. for C_32_H_28_F_2_O_2_Na^+^ [M + Na]^+^ 505.1950, found 505.1951.

*5,5*′*-Diallyl-2-((3-fluorobenzyl)oxy)-1,1*′*-biphenyl-2*′*-ol (***5a***).* Yield: 65.9%, pale yellow oils; ^1^H-NMR (DMSO-*d_6_*, 500 MHz): δ 9.08 (1H, s, –OH), 7.34 (1H, m, H-15), 7.14 (2H, m, H-6, H-6′), 7.07 (2H, m, H-4, H-16), 7.01 (2H, m, H-3, H-14), 6.95 (2H, m, H-3′, H-4′), 6.84 (1H, dd, *J* = 8.1, 3.4 Hz, H-12), 5.94 (2H, m, H-8, H-8′), 5.09 (2H, m, H-9), 5.06 (2H, s, H-10), 5.04 (2H, m, H-9′), 3.33 (2H, d, *J* = 6.8 Hz, H-7), 3.27 (2H, d, *J* = 6.7 Hz, H-7′); ^13^C NMR (DMSO-*d_6_*, 125 MHz): δ 161.2 (d, ^1^*J*_CF_ = 241.6 Hz, C-13), 153.8 (C-2′), 153.0 (C-2), 140.5 (d, ^3^*J*_CF_ = 7.5 Hz, C-11), 138.2 (C-8′), 138.0 (C-8), 131.5 (C-5′), 131.4 (C-5), 131.2 (C-6′), 130.1 (d, ^3^*J*_CF_ = 8.2 Hz, C-15), 129.3 (C-6), 128.2 (C-4′), 128.1 (C-4), 128.0 (C-1), 125.2 (C-1′), 122.8 (d, ^4^*J*_CF_ = 2.7 Hz, C-16), 115.5 (C-3′), 115.3 (C-9′), 115.2 (C-9), 114.2 (d, ^2^*J*_CF_ = 20.8 Hz, C-14), 113.6 (d, ^2^*J*_CF_ = 21.8 Hz, C-12), 112.8 (C-3), 68.5 (C-10), 38.7 (C-7, C-7′); HRESIMS: calcd. for C_25_H_23_FO_2_Na^+^ [M + Na]^+^ 397.1574, found 397.1575.

*5,5*′*-Diallyl-2,2*′*-bis ((3-fluorobenzyl)oxy)-1,1*′*-biphenyl (***5b***).* Yield: 2.8%, pale yellow oils; ^1^H-NMR (DMSO-*d_6_*, 500 MHz): δ 7.30 (2H, m, H-15, H-15′), 7.12 (2H, dd, *J* = 8.35, 2.25 Hz, H-4, H-4′), 7.07 (2H, s, H-6, H-6′), 7.06 (6H, m, Ar-H), 7.00 (2H, d, *J* = 10.1 Hz, H-3, H-3′), 5.94 (2H, m, H-8, H-8′), 5.03 (4H, m, H-9, H-9′), 5.03 (4H, s, H-10, H-10′), 3.32 (4H, d, *J* = 6.7 Hz, H-7, H-7′); ^13^C NMR (DMSO-*d_6_*, 125 MHz): δ 161.1 (d, ^1^*J*_CF_ = 241.9 Hz, C-13, C-13′), 153.7 (C-2, C-2′), 140.4 (d, ^3^*J*_CF_ = 7.4 Hz, C-11, C-11′), 137.9 (C-8, C-8′), 131.6 (C-5, C-5′), 131.4 (C-6, C-6′), 130.2 (d, ^3^*J*_CF_ = 8.1 Hz, C-15, C-15′), 128.4 (C-4, C-4′), 127.5 (C-1, C-1′), 122.7 (d, ^4^*J*_CF_ = 2.5 Hz, C-16, C-16′), 115.6 (C-9, C-9′), 114.2 (d, ^2^*J*_CF_ = 20.8 Hz, C-14, C-14′), 113.5 (d, ^2^*J*_CF_ = 21.9 Hz, C-12, C-12′), 112.8 (C-3, C-3′), 68.5 (C-10, C-10′), 38.6 (C-7, C-7′); HRESIMS: calcd. for C_32_H_28_F_2_O_2_Na^+^ [M + Na]^+^ 505.1950, found 505.1953.

*5,5*′*-Diallyl-2-((4-fluorobenzyl)oxy)-1,1*′*-biphenyl-2*′*-ol (***6a***).* Yield: 34.7%, pale yellow oils; ^1^H-NMR (DMSO-*d_6_*, 500 MHz): δ 9.04 (1H, s, –OH), 7.37 (2H, dd, *J* = 8.3, 5.6 Hz, H-12, H-16), 7.12 (2H, m, H-13, H-15), 7.09 (1H, dd, *J* = 8.4, 2.3 Hz, H-4), 7.02 (2H, d, *J* = 8.4 Hz, H-3, H-3′), 6.94 (2H, m, H-6, H-6′), 6.83 (1H, m, H-4′), 5.93 (2H, m, H-8, H-8′), 5.05 (4H, m, H-9, H-9′), 5.01 (2H, s, H-10), 3.32 (2H, d, *J* = 6.8 Hz, H-7), 3.26 (2H, d, *J* = 6.8 Hz, H-7′); ^13^C NMR (DMSO-*d_6_*, 125 MHz): δ 160.5 (d, ^1^*J*_CF_ = 241.7 Hz, C-14), 153.9 (C-2′), 152.9 (C-2), 138.2 (C-8′), 138.0 (C-8), 133.7 (d, ^4^*J*_CF_ = 2.9 Hz, C-11), 131.5 (C-5′), 131.4 (C-5), 131.2 (C-6′), 129.3 (C-6), 129.2 (d, ^3^*J*_CF_ = 8.1 Hz, C-12, C-16), 128.2 (C-4′), 128.1 (C-4), 127.9 (C-1), 125.3 (C-1′), 115.5 (C-3′), 115.3 (C-9′), 115.2 (C-9), 115.0 (d, ^2^*J*_CF_ = 21.2 Hz, C-13, C-15), 113.0 (C-3), 68.8 (C-10), 38.7 (C-7, C-7′); HRESIMS: calcd. for C_25_H_23_FO_2_Na^+^ [M + Na]^+^ 397.1574, found 397.1575.

*5,5*′*-Diallyl-2,2*′*-bis ((4-fluorobenzyl)oxy)-1,1*′*-biphenyl (***6b***).* Yield: 4.7%, rhombic white crystal; ^1^H-NMR (DMSO-*d_6_*, 500 MHz): δ 7.26 (4H, m, Ar-H), 7.07 (4H, m, Ar-H), 7.06 (2H, m, H-4, H-4′), 7.04 (2H, m, H-3, H-3′), 7.03 (2H, s, H-6, H-6′), 5.93 (2H, m, H-8, H-8′), 5.04 (4H, m, H-9, H-9′), 5.00 (4H, s, H-10, H-10′), 3.31 (4H, d, *J* = 6.8 Hz, H-7, H-7′); ^13^C NMR (DMSO-*d_6_*, 125 MHz): δ 160.5 (d, ^1^*J*_CF_ = 241.7 Hz, C-14, C-14′), 153.8 (C-2, C-2′), 138.0 (C-8, C-8′), 133.6 (d, ^4^*J*_CF_ = 2.9 Hz, C-11, C-11′), 131.5 (C-5, C-5′), 131.4 (C-6, C-6′), 129.1 (d, ^3^*J*_CF_ = 8.1 Hz, C-12, C-16, C-12′, C-16′), 128.3 (C-4, C-4′), 127.5 (C-1, C-1′), 115.5 (C-9, C-9′), 115.0 (d, ^2^*J*_CF_ = 21.2 Hz, C-13, C-15, C-13′, C-15′), 112.9 (C-3, C-3′), 68.7 (C-10, C-10′), 38.6 (C-7, C-7′); HRESIMS: calcd. for C_32_H_28_F_2_O_2_Na^+^ [M + Na]^+^ 505.1950, found 505.1952.

*5,5*′*-Diallyl-2-((2-chlorobenzyl)oxy)-1,1*′*-biphenyl-2*′*-ol (***7a***).* Yield: 42.6%, pale yellow oils; ^1^H-NMR (DMSO-*d_6_*, 500 MHz): δ 9.05 (1H, s, –OH), 7.44 (2H, m, H-6, H-6′), 7.31 (1H, m, H-13), 7.26 (1H, m, H-14), 7.10 (1H, dd, *J* = 8.3, 2.3 Hz, H-4), 7.03 (2H, m, H-15, H-16), 6.94 (2H, m, H-3, H-4′), 6.81 (1H, d, *J* = 8.1 Hz, H-3′), 5.92 (2H, m, H-8, H-8′), 5.08 (2H, s, H-10), 5.03 (4H, m, H-9, H-9′), 3.33 (2H, d, *J* = 6.9 Hz, H-7), 3.25 (2H, d, *J* = 6.7 Hz, H-7′); ^13^C NMR (DMSO-*d_6_*, 125 MHz): δ 153.7 (C-2′), 152.9 (C-2), 138.2 (C-8′), 138.0 (C-8), 134.8 (C-11), 131.7 (C-5′), 131.5 (C-5, C-6′), 131.3 (C-6), 129.3 (C-12), 129.2 (C-4′), 129.0 (C-13, C-14), 128.2 (C-4), 128.1 (C-16), 128.0 (C-1), 127.1 (C-1′), 125.0 (C-15), 115.5 (C-9′), 115.3 (C-3′), 115.2 (C-9), 112.9 (C-3), 66.9 (C-10), 38.7 (C-7, C-7′); HRESIMS: calcd. for C_25_H_23_ClO_2_Na^+^ [M + Na]^+^ 413.1279, found 413.1280.

*5,5*′*-Diallyl-2-((3-chlorobenzyl)oxy)-1,1*′*-biphenyl-2*′*-ol (***8a***).* Yield: 37.7%, pale yellow oils; ^1^H-NMR (DMSO-*d_6_*, 500 MHz): δ 9.07 (1H, s, –OH), 7.38 (1H, s, H-12), 7.32 (3H, m, H-14, H-6, H-6′), 7.09 (1H, dd, *J* = 8.3, 2.3 Hz, H-4), 7.02 (2H, m, H-15, H-16), 6.95 (2H, m, H-3, H-4′), 6.84 (1H, d, *J* = 8.2 Hz, H-3′), 5.94 (2H, m, H-8, H-8′), 5.05 (2H, s, H-10), 5.04 (4H, m, H-9, H-9′), 3.33 (2H, d, *J* = 6.8 Hz, H-7), 3.28 (2H, d, *J* = 6.8 Hz, H-7′); ^13^C NMR (DMSO-*d_6_*, 125 MHz): δ 153.7 (C-2′), 152.9 (C-2), 140.2 (C-11), 138.2 (C-8′), 138.0 (C-8), 133.0 (C-13), 131.6 (C-5′), 131.4 (C-5), 131.2 (C-6′), 130.0 (C-6), 129.3 (C-15), 128.3 (C-14), 128.1 (C-4′), 128.0 (C-4), 127.3 (C-1), 126.7 (C-1′), 125.4 (C-12), 125.2 (C-16), 115.5 (C-9′), 115.4 (C-3′), 115.2 (C-9), 113.2 (C-3), 68.5 (C-10), 38.7 (C-7, C-7′); HRESIMS: calcd. for C_25_H_23_ClO_2_Na^+^ [M + Na]^+^ 413.1279, found 413.1281.

*5,5*′*-Diallyl-2-((4-chlorobenzyl)oxy)-1,1*′*-biphenyl-2*′*-ol (***9a***).* Yield: 37.7%, pale yellow oils; ^1^H-NMR (DMSO-*d_6_*, 500 MHz): δ 9.04 (1H, s, –OH), 7.35 (4H, m, H-6, H-13, H-15, H-6′), 7.08 (1H, dd, *J* = 8.5, 2.3 Hz, H-4), 7.01 (2H, m, H-12, H-16), 6.93 (2H, m, H-3, H-4′), 6.82 (1H, d, *J* = 8.1 Hz, H-3′), 5.93 (2H, m, H-8, H-8′), 5.08 (2H, m, H-9), 5.03 (2H, s, H-10), 4.99 (2H, m, H-9′), 3.32 (2H, d, *J* = 6.8 Hz, H-7), 3.27 (2H, d, *J* = 6.8 Hz, H-7′); ^13^C NMR (DMSO-*d_6_*, 125 MHz): δ 153.8 (C-2′), 152.9 (C-2), 138.3 (C-8′), 138.0 (C-8), 136.6 (C-11), 132.0 (C-5′), 131.5 (C-14), 131.4 (C-5), 131.2 (C-6′), 129.3 (C-6), 128.9 (C-12, C-16), 128.2 (C-4′), 128.1 (C-4, C-13, C-15), 127.9 (C-1), 125.2 (C-1′), 115.5 (C-9′), 115.3 (C-3′), 115.2 (C-9), 112.9 (C-3), 68.7 (C-10), 38.7 (C-7, C-7′); HRESIMS: calcd. for C_25_H_23_ClO_2_Na^+^ [M + Na]^+^ 413.1279, found 413.1280.

### 3.3. MTT Assay

Human breast cancer (MDA-MB-231 and MCF-7), nasopharyngeal carcinoma (CNE-2Z), and hepatocellular carcinoma (SMMC-7721) cell lines were cultured in RPMI-1640 medium or DMEM that were supplemented with 10% fetal bovine serum and 1% penicillin-streptomycin in a 5% CO_2_ incubator at 37 °C. All cells were seeded in 96-well plates at a density of 5000 cells per well and treated with various concentrations (0, 6.25, 12.5, 25, and 50 μM) of the derivatives for 72 h. Taxol was used as a positive control. Cytotoxic activity was evaluated using standard MTT assay procedures as previously described. (All samples were lyophilized prior to MTT assay because of solvent contaminants peaks in the NMR spectra)

### 3.4. Wound-Healing Assay

MDA-MB-231 cells were seeded in 6-well plates at a density of 6 × 10^5^ cells per well. The monolayered cells were wounded by scratching with 100 μL pipette tips, then washed with phosphate-buffered saline (PBS). The PBS was then replaced with serum-free RPMI-1640 containing the compound of interest. The images were taken at 0 h and 24 h after incubation for 24 h at 37 °C. The wound assay values were obtained from three randomly selected fields. Similar patterns of inhibition were observed in three independent experiments.

### 3.5. Cell Migration Assay

Cell migration assays were performed using a 24-well plate with 8.0 μm pore membrane inserts (Corning, NY, USA) without Matrigel. MDA-MB-231 cells were added to the upper chambers at a concentration of 2.5 × 10^5^ cells per well and incubated for 24 h after treatment with various concentrations (0, 5, 10, and 20 μM) of **6a**. The lower chambers were filled with conditioned media. After 24 h, the cells that had migrated were stained with 0.1% crystal violet and photographed under a light microscope at 200× magnification. Taxol was used as a positive control.

### 3.6. Cell Invasion Assay

Cell invasion assays were performed using a 24-well plate with 8.0 μm pore membrane inserts that were coated with 50 μL of Matrigel (BD, Franklin Lakes, NJ, USA) and incubated at 37 °C for 1.0 h. MDA-MB-231 cells (3 × 10^5^ cells per well) were added to the upper chambers and incubated with various concentrations (0, 5, 10, and 20 μM) of **6a** for 36 h. The rest of the process was the same as that described in the Section 2.5.

### 3.7. Western Blotting

MDA-MB-231 cells were cultured in 6-well plates at a density of 3 × 10^5^ cells per well. After adherence, the cells were treated with various concentrations (0, 10, 20, and 40 μM) of **6a** and incubated for 24 h. The cells were harvested, washed with PBS, and proteins were extracted and quantified. The proteins were separated by SDS-PAGE and PVDF membranes. The PVDF membrane was blocked with 5% skim milk and incubated with primary antibodies (HIF-1α, MMP-2, and MMP-9) at 4 °C overnight. After washing with tris-buffered saline/tween 20 buffer, the membranes were incubated with the corresponding secondary antibodies at room temperature for 2.0 h. Protein bands were visualized using a chemiluminescence kit and detected using a gel imaging system (Bio-Rad, Hercules, CA, USA). Anti-β-actin was used as an internal control.

### 3.8. Statiscal Analysis

Statistical analysis was performed with two samples using SPSS 16.0 software (Armonk, NY, USA), and * *p* < 0.05 or ** *p* < 0.01 were considered statistically significant differences.

## 4. Conclusions

In summary, a series of magnolol derivatives **1a**–**9a** and **1b**–**6b** were semisynthesized by Williamson reaction and evaluated for their in vitro antiproliferative activity by MTT assay on four different human cancer cell lines (MDA-MB-231, MCF-7, CNE-2Z, and SMMC-7721). The results showed that most of the magnolol derivatives exhibited better in vitro antiproliferative activity than the precursor magnolol. Among them, compound 6a had the best cytotoxic activity against MDA-MB-231 cells with an IC_50_ value of 20.43 μM. Preliminary SAR indicated that *O*-alkylation of magnolol at 2-OH or 2**′**-OH could enhance the cytotoxic activity with the benzyl with F-substituted has better activity. In addition, the results of wound-healing and transwell assays showed that 6a could also inhibit the migration of MDA-MB-231 cells very well. A more detailed mechanistic study demonstrated that 6a inhibited the migration and invasion of MDA-MB-231 cells by downregulating HIF-1α, MMP-2, and MMP-9 protein levels. Our findings will give some basis to the development of magnolol derivatives as potential anti-cancer candidates for the treatment of human cancer.

## Data Availability

The data presented in this study are available on request from the corresponding authors.

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
