# Peer review of "Semi-Synthesis and In Vitro Anti-Cancer Evaluation of Magnolol Derivatives"

_molecules, 2021, doi:10.3390/molecules26144302_

Round 1

Reviewer 1 Report

Please refer to the observations/suggestions available as notes in the enclosed files (main document and supplementary materials).

Reviewer 2 Report

Please find my comments below:

  • I don’t see the Scheme 1. Please add this picture.
  • The words like “in vitro” should be written italic font.
  • NMR: The compounds like 4a or 4b are not pure. Have all compounds of this purity been bio-tested? If so, the results are not reliable.

Reviewer 3 Report

The manuscript entitled “Semi-synthesis and in vitro anti-cancer evaluation of magnolol derivatives” presented by Xiaolong Sun and co-authors describes study of anti-cancer  potential of natural compounds. The authors have tested effects of magnolol derivatives.

This work is adequate written and prepared and should be accepted for publication after major revision.

Abstract:

-ln 14: “Of these derivatives” that doesn’t mean what you wanted to say. Correct it for example like this “From these derivatives”.

Also the title has in vitro written normal, it should have been in italic. Correct that trough entire document.

Introduction:

Should be written in more details.

Results and Discussion: 

-The whole discussion section is missing in this paper. It is necessary to compare your results with published results for similar compounds.

-Scheme 1 is missing

Owerall figures look good, and the results are fine. This will be one good publication, but please write more in introduction and conclusion, and discus the results with some results previously published for some similar natural compounds.   

Reviewer 4 Report

This manuscript (ID:  molecules-1284542) describes a family of compounds related to the natural product magnolol which has shown modest activity against several human cancers.  As I am not familiar with the cell lines mentioned in the paper, it would be nice if the authors indicated which types of cancers are being studied.  It is indirectly mentioned that MDA-MB-231 are breast cancer cells, but are they all breast cancer cells?  The compounds themselves are not very active but compound 6a does appear to suppress invasion and metastasis.  There is a lot of biology in this paper, and since I am a chemist, I will let the other reviewers comment in this area.  The paper is relatively well written, though I have included a few things that the authors could clarify below.  Otherwise, I noted in the Supplemental Material that many of the compound spectra contain small amounts of N,N-dimethylformamide (1a, 2b, 4a, 4b) and others (3a, 4a, 5a-9a) have a significant amount of ethyl acetate in them.  It is not clear if all of these contaminants were removed before biological testing.  The spectra of 2a and 5a are not clear because they are too dilute.

Clarification of English:

Abstract:  In addition, western blotting experiments, performed using various concentrations of 6a, demonstrated...

p 1

line 30:  In contrast to chemotherapy,...  (delete with)

line 34:  Natural products can have many advantages...  (some natural products are quite toxic)

p 2

five lines from bottom:  in vitro   (italic)

three lines from the bottom:  Were the IC50's calculated or determined?  (determined is better)

p 7

line 150:  N,N-dimethylformamide  (N's italic and one word)

Yields are not normally reported to two decimal places.

There are no fluorine coupling constants in the 13C NMRs of 5a,b or 6a,b

Compounds 4a, 4b, 5a, 5b, 6a, 7a and 8a should be described as "pale yellow oils" rather than

   "pale yellow oily"

p 9, line 285:  ...1% penicillin-streptomycin in a 5% CO2 incubator...

p 9, line 294:  0 h and 24 h

If the authors will address these issues, the paper would be acceptable for publication.

Round 2

Reviewer 1 Report

The authors addressed all the previous observations and the manuscript has been deeply improved.

I've only one single suggestion:

The authors should check if there is any format to report the NMR data in the journal guidelines.

Generally these data are reported in the following order: chemical shift value in ppm or delta, in brackets integral value, multiplicity, coupling constant(s) value(s) in Hz, assigned nucleus/nuclei.

e.g.: 7.13 (1H, dd,  J = 7.0, 1.9 Hz, H-4)...

as these are now in the manuscript the integrals values and the multiplicity are reversed.

Author Response

Dear reviewers and editors,

Thank you very much for writing again to inform us about our paper entitled “Semi-synthesis and in vitro anti-cancer evaluation of magnolol” (Manuscript ID: molecules-1284542). We appreciated for your effort, and we have marked up all the altered passages with "Track Changes" function. The responses and revisions are summarized as follows.

Point 1: The authors should check if there is any format to report the NMR data in the journal guidelines. Generally these data are reported in the following order: chemical shift value in ppm or delta, in brackets integral value, multiplicity, coupling constant(s) value(s) in Hz, assigned nucleus/nuclei.

e.g.: 7.13 (1H, dd, J = 7.0, 1.9 Hz, H-4)...

Response 1: We have modified the NMR data according to the format of "7.13 (1H, dd, J = 7.0, 1.9 Hz, H-4)...".

Reviewer 3 Report

Thank you for the corrections you made, in this form your work can be published in this journal. 

Author Response

Thank you!

Reviewer 4 Report

I am still concerned by the solvent contaminants (up to 5-10%) in the spectra of the compounds (DMF contaminant-1a, 2a, 4a, 4b) and (ethyl acetate contaminant-3a, 4a, 5a-9a).  The authors did not address this issue in their response to my comments.  If these contaminants were not removed, they could have affected the observed activity of the compounds.  Some comment should be included if further efforts were made to remove these solvents prior to biological testing, even if it is in the SI.  Most of my other concerns have been addressed.  I will point out two errors I noted in the text:

line 122:  a space is needed between "regulator" and "is"

line 174:  N,N-dimethylformamide is one word

Author Response

Dear reviewers and editors,

Thank you very much for writing again to inform us about our paper entitled “Semi-synthesis and in vitro anti-cancer evaluation of magnolol” (Manuscript ID: molecules-1284542). We appreciated for your effort, and we have marked up all the altered passages with "Track Changes" function. The responses and revisions are summarized as follows.

Point 1: I am still concerned by the solvent contaminants (up to 5-10%) in the spectra of the compounds (DMF contaminant-1a, 2a, 4a, 4b) and (ethyl acetate contaminant-3a, 4a, 5a-9a). The authors did not address this issue in their response to my comments. If these contaminants were not removed, they could have affected the observed activity of the compounds. Some comment should be included if further efforts were made to remove these solvents prior to biological testing, even if it is in the SI. 

Response 1:

(1) We have clearly checked the spectra. The peaks of the solvent contaminants (DMF and EtOAc) are indicated in the spectra of the supplemental material.

(2) We have lyophilized the samples for the activity test, and we guarantee that all samples are over 95% pure and correct for 100% purity at the time of drug configuration. In addition, the sentence, "All samples were lyophilized prior to MTT assay because of solvent contaminants peaks in the NMR spectra", is added at the section "3.3. MTT assay".

(3) We apologize for not responding to your the comment in a timely manner.

Point 2: line 122: a space is needed between "regulator" and "is"

Response 2: Done.

Point 3: line 174: N,N-dimethylformamide is one word

Response 3: Done.